# Research on Safety Helmet Detection Algorithm Based on Improved YOLOv5s

**DOI:** 10.3390/s23135824

**Published:** 2023-06-22

**Authors:** Qing An, Yingjian Xu, Jun Yu, Miao Tang, Tingting Liu, Feihong Xu

**Affiliations:** 1School of Artificial Intelligence, Wuchang University of Technology, Wuhan 430223, China; 120160450@wut.edu.cn (Q.A.); 120160311@wut.edu.cn (M.T.); 120160804@wut.edu.cn (T.L.); 120160287@wut.edu.cn (F.X.); 2School of Safety Science and Emergency Management, Wuhan University of Technology, Wuhan 430079, China; 3USTC iFLYTEK Co., Ltd., Hefei 230088, China; junyu@iflytek.com

**Keywords:** detection, YOLOv5, SIoU, combinatorial attention mechanisms, K-means++, knowledge distillation

## Abstract

Safety helmets are essential in various indoor and outdoor workplaces, such as metallurgical high-temperature operations and high-rise building construction, to avoid injuries and ensure safety in production. However, manual supervision is costly and prone to lack of enforcement and interference from other human factors. Moreover, small target object detection frequently lacks precision. Improving safety helmets based on the helmet detection algorithm can address these issues and is a promising approach. In this study, we proposed a modified version of the YOLOv5s network, a lightweight deep learning-based object identification network model. The proposed model extends the YOLOv5s network model and enhances its performance by recalculating the prediction frames, utilizing the IoU metric for clustering, and modifying the anchor frames with the K-means++ method. The global attention mechanism (GAM) and the convolutional block attention module (CBAM) were added to the YOLOv5s network to improve its backbone and neck networks. By minimizing information feature loss and enhancing the representation of global interactions, these attention processes enhance deep learning neural networks’ capacity for feature extraction. Furthermore, the CBAM is integrated into the CSP module to improve target feature extraction while minimizing computation for model operation. In order to significantly increase the efficiency and precision of the prediction box regression, the proposed model additionally makes use of the most recent SIoU (SCYLLA-IoU LOSS) as the bounding box loss function. Based on the improved YOLOv5s model, knowledge distillation technology is leveraged to realize the light weight of the network model, thereby reducing the computational workload of the model and improving the detection speed to meet the needs of real-time monitoring. The experimental results demonstrate that the proposed model outperforms the original YOLOv5s network model in terms of accuracy (Precision), recall rate (Recall), and mean average precision (mAP). The proposed model may more effectively identify helmet use in low-light situations and at a variety of distances.

## 1. Introduction

Construction workers frequently incur injuries due to their failure to wear safety helmets [1]. Based on accident data from 2015 to 2018, out of 78 construction-related accidents, 53 incidents, or 67.95 percent of all accidents, were brought on by failure to wear safety helmets [2]. Consequently, it is crucial to verify that safety helmets are being. Currently, detecting whether employees are wearing safety helmets depends on human monitoring, which has the disadvantages of being expensive and failing to provide real-time detection [3]. Numerous image-based detection methods have been proposed to address this issue, both conventional image detection techniques and methods for detecting objects using deep learning technology [4,5].

The Viola–Jones algorithm, histograms of oriented gradients (HOG) + support vector machine (SVM), and others are examples of traditional image detection techniques for safety helmet identification and the development of deformable parts models (DPM). Viola–Jones (VJ) can process targets in real time with high accuracy [6]. This method comprises three structures, namely, the feature type and evolution, learning algorithm, and cascade structure. Moreover, it requires a high detection rate for a single classifier, such that each classifier must obtain a detection rate of 99.7% in order to attain an overall detection rate of 90% [7]. Some researchers conducted a detailed analysis of feature-oriented classification storage and a feature matching query utilizing HOG feature extraction and SVM to improve target recognition and classification efficiency while considering the different granularity displayed by the test images [8]. DPM is an extension of HOG that trains a gradient model of the object using SVM, matches the model with the target, and achieves target classification. In the early stages of image recognition, conventional target detection methods required the extraction of a large number of target features, which had two main disadvantages: a large number of logical operations were needed to generate sufficient candidate regions, and the complexity of the characteristics prevented the identification speed and accuracy from meeting the objectives [9].

Girshick et al. introduced a target detection method that utilized a deep convolutional neural network, which effectively overcame the limitations of existing target detection technologies. Deep learning target identification approaches now fall into two main categories: one-stage methods based on regression and two-stage methods based on candidate region selection [10,11,12]. Girshick’s region-convolutional neural network (R-CNN) is one of the two-stage techniques used to extract picture information [13]. However, R-CNN faces difficulties when generating candidate frames in complex backgrounds, and scaling and cropping during feature extraction may result in the loss of image information. In [14], Ross et al. proposed the famous fast region with CNN (Fast R-CNN) network, which replaces the spatial pooling layer of SPP-Net, simplifying the network model and saving computing resources. However, region pruning relies on a selective search method to generate regions of interest and cannot be accelerated by GPU. The same year saw the introduction of faster regions with CNN (Faster R-CNN) by Renetal et al., which leverages a region prediction network (RPN) to replace the conventional region prediction algorithm and uses a fully connected layer to enhance image robustness [15]. However, Faster R-CNN cannot share the parameters of multiple related regions in the second stage, which increases computational overhead. In addition, using the fully connected layer may lead to information loss [16,17].

The single shot detector (SSD) and you only look once (YOLO) algorithms are part of the one-stage target identification algorithm, which is a quicker technique. Redmon et al. introduced the YOLO [18] model, which transforms the challenging two-step detection procedure into an abstract regression issue. A multi-scale-based detection technique called SSD, which can effectively find several small objects, was proposed by Liu et al. [19]. However, the preprocessing of minor things could be optimized after the SSD algorithm performs depth convolution. YOLOv2, developed by Redmon et al., employs DarkNet-19 as a novel fundamental model and permits end-to-end training [20]. They also presented the YOLOv3 [21] network, which considerably improves the model’s ability to recognize objects of varying sizes by fusing three feature layers of different sizes using feature pyramid network (FPN) technology. Park et al. proposed two-step real-time night-time fire detection in urban environments using Static ELASTIC-YOLOv3 [22]. YOLOv4 was introduced by Bochkovski et al. and leverages the Cross-Stage Partial (CSP) Darknet-53 as the backbone network and replaces the FPN algorithm in the YOLOv3 network [23]. As a result, the model’s detection precision was significantly increased. Lin et al. proposed utilizing the improved YOLOv4 to perform defect detection on stitched images of rotating tools, and it has demonstrated good performance [24]. YOLOv5 is based on YOLOv4, which was proposed by Glenn [25]. YOLOv4 introduced a focus module to boost detection speed and accuracy, and it is now a model with excellent target recognition accuracy. Wang et al. proposed an improved YOLOv5 cotton foreign fiber detection and classification based on polarization imaging, which improved the accuracy and speed of detection [26]. Based on the YOLOv5 model, this research suggests an enhanced YOLOv5s hard hat detection approach. The prediction box was first adjusted by strengthening the YOLOv5 adaptive anchor. In order to improve the feature extraction of small targets and, hence, increase the detection accuracy of the targets, we incorporated a combined attention method. Finally, we improved the regression box’s accuracy by using SIoU Loss as the bounding box loss function. The experiments the authors conducted with their hard hat dataset show that the enhanced YOLOv5s model can effectively extract characteristics from small targets. The approach put forth in this research can be used in the field of helmet detection to reduce worker fatalities.

## 2. Proposed Method

The technical roadmap of this paper is shown in Figure 1.

In order to further select the most accurate algorithm, the actual detection effect of the current mainstream target detection algorithm model was utilized in real images of small targets with complex backgrounds, as shown in Figure 2. In Figure 2a, we show the actual detection effect of the SSD model, where the confidence of person is 0.69, and the confidence of hat is 0.70. Figure 2b shows the actual detection effect of the Fast R-CNN model Figure 1, where the confidence of person is 0.65, and the confidence of hat is 0.68. Figure 2c is the actual detection effect diagram of the Faster R-CNN model, where the confidence of person is 0.71, and the confidence of hat equals 0.72. Figure 2d demonstrates the actual detection effect diagram of the YOLOv5s model, where the confidence of person is 0.73, and the confidence of hat is 0.73. Compared with other mainstream target detection algorithms, the experimental results illustrate that the YOLOv5s target detection algorithm has a higher detection accuracy than the SSD model, Fast R-CNN model, and Faster R-CNN model, while maintaining a light weight.

In order to create an object detection algorithm that can effectively detect small objects, especially helmets, this paper proposes an improved YOLOv5s network model based on the YOLOv5s model. This method can effectively enhance the ability to extract helmet target features.

A real-time target identification system called YOLOv5 offers four network models with varying degrees of depth: YOLOv5s, YOLOv5m, YOLOv5l, and YOLOv5x. The lightweight YOLOv5s network structure, shown in Figure 3, is made up of four parts: input (Input), backbone network (Backbone), neck network (Neck), and detecting head (Prediction). This research focuses on enhancing this structure. The focus module, the CBL convolutional layer, and the CSP1_X module are the components of the YOLOv5s backbone network. A 640 × 640 × 3 picture is fed into the focus structure, followed by slice processing, and a convolution operation yields a 320 × 320 × 64 feature map. The CBL convolutional layer and the CSP1_X module are then used to create a rich feature map with semantic information. The neck network implements two upsampling operations using CSP2_X and FPN+PAN models to combine shallow and high-level semantic features, realizing the fusion of multi-scale receptive fields and enhancing the feature fusion ability. For the prediction, we used the regression + classification method, dividing the input image into three different sized grids: 80 × 80, 40 × 40, and 20 × 20, thereby identifying large, medium, and small targets. Furthermore, YOLOv5 applies adaptive picture scaling, adaptive anchor frame computation, and mosaic data improvement to the input. The backbone network receives the focus and CSP structures, whereas the neck network receives the FPN+PAN structure [27]. The target detection frame in the output terminal employs GIoU_Loss as its loss function. We also suggest the NMS non-maximum suppression approach. The YOLOv5s algorithm not only increases detection accuracy when compared to the conventional two-stage detection approach, but also significantly reduces training time.

The optimization of the YOLOv5s algorithm for helmet detection can be divided into several aspects:(1)Anchor box adjustment: The anchor box is adjusted using the K-means++ algorithm to place it closer to the actual target box, thereby improving the initial preselection box.(2)Network structure improvement: An attention mechanism is incorporated to optimize the network topology, increasing the model’s effectiveness and precision.(3)Bounding box loss function optimization: The prediction box regression is made faster and more accurate by optimizing the loss function for the bounding box at the output end.(4)The knowledge distillation technology is used to realize the light weight of the network model, thereby reducing the computational workload of the model, increasing the detection speed, and meeting the needs of real-time monitoring.

### 2.1. Improvement of Adaptive Anchor Frame Mechanism in YOLOv5 Based on K-means++ Algorithm

A core problem in computer vision is object recognition, which entails locating and recognizing items inside an image using bounding boxes. To increase the object recognition models’ accuracy, selecting an appropriate prior bounding box during training can be beneficial. The YOLOv5 model incorporates the concept of an anchor box into target recognition. An initial bounding box with a defined size and aspect ratio is known as an anchor box. The anchor box’s proximity to the ground-truth bounding box is taken into account by the model when adjusting the predicted bounding box during training. K-means and genetic algorithms were employed to update the anchor boxes in the initial YOLOv5 model [28], with the Euclidean distance acting as the metric function. However, when working with samples of different sizes, utilizing Euclidean distance may result in clustering problems. To solve this problem, we suggest a hybrid method that groups anchor boxes using the K-means++ algorithm and the intersection-over-union (IOU) distance metric. This results in previous bounding boxes with a higher *IOU* value, improving object recognition accuracy.

The dimensions of the two boxes are represented by (*w*_1_, *h*_1_) and (*w*_2_, *h*_2_), respectively, as illustrated in Figure 4. The region highlighted in red denotes the intersection of the two boxes, with dimensions (*w*, *h*), and is defined as:(1)$$S_{j}=w\times h.$$

The combined area of the two frames is represented by the “blue + red + gray” region. This combined area can be calculated using a formula, which is rewritten as:(2)$$S_{b}=w1\times h1+w2\times h2-w\times h,$$

The *IOU* can be obtained based on Equations (1) and (2):(3)$$IOU=\frac{S_{j}}{S_{b}}$$

The overlap between two frames is measured by the *IOU*, a statistic that has a scale from 0 to 1. There is no gap between the two frames when the value is 0, while a value of 1 indicates that the two frames are identical. When the *IOU* value is higher, it indicates that the two previous frames fit better. To ensure that the measurement value and similarity have a negative correlation, when the measurement value is low, the similarity is high, and the value of the *IOU* is subtracted from 1. This gives rise to Equation (4), which calculates the similarity metric between two frames:(4)$$d_{iou}=1-IOU$$

In this paper, we utilized the K-means-based YOLOv5 algorithm in combination with the Euclidean distance measure method to derive 9 prior boxes. These prior boxes corresponded to feature maps of varying scales and had a matching degree of 0.8553. The prior boxes on feature maps of different scales are presented in Table 1.

The linear growth in the computing complexity and quick convergence time of the K-means clustering method are two of its many benefits. The beginning clustering center must be predetermined for this approach, and various initial clustering centers may provide different clustering outcomes. To address this problem, we leveraged the K-means++ technique to calculate the anchor boxes in our object identification model. The first cluster center is chosen at random by the K-means++ algorithm, ensuring that the mutual distance between the initial cluster centers is as great as is feasible. When the initial n cluster centers (0 n K) have been chosen, the n + 1-th cluster center is chosen by giving sites further from the n cluster centers a greater likelihood. This approach helps to ensure that the anchor boxes are optimized for better accuracy and robustness in object recognition while mitigating the potential effects of initial clustering center selection.

To create previous boxes of feature maps with various scales for this investigation, we combined the IOU measurement method with the K-means++ algorithm. Our approach yielded a matching degree of 0.8689 for the previous frames, which was higher than that achieved by clustering with the K-means algorithm. The resulting prior box distribution is presented in Table 2.

### 2.2. Improvement of Network Structure

The attention mechanism seeks to identify relevant information and disregard irrelevant information, thereby enhancing the efficiency of neural networks. By obtaining detailed information and suppressing unnecessary data, it becomes possible to improve the network’s performance [29,30]. In order to do this, we suggest a fusion approach that combines the cross-stage partial (CSP) module built into the convolutional block attention module (CBAM) attention mechanism with the global attentional map (GAM) mechanism. Our method attempts to improve the model’s overall performance by strengthening its feature extraction capabilities.

(1)CBAM attention mechanism

A compact and adaptable module for strengthening neural networks is the convolutional block attention module (CBAM) [31]. In this study, the last layer of the cross-stage partial (CSP) modules in the backbone and neck of YOLOv5s includes the CBAM module. This integration enhances the model’s ability to extract features while also lowering computational complexity.

The channel attention module and the spatial attention module are two sub-modules that make up the CBAM module. They are used in succession. From the deep network, we first achieve intermediate feature maps. The CBAM modules are then used at each convolutional block to adaptively improve these maps. The attention map is then successively inferred along the channel and space dimensions. To accomplish adaptive feature refinement, the output attention map is multiplied by the input feature map. In Figure 5, we can observe the detailed CBAM attention module of the proposed method.

The intermediate feature map F∈R*^C^*^×*H*×*W*^ is the input for the CBAM module. A 1D channel attention map (M*_c_*∈R*^C^^×^*^1^*^×^*^1^) and a 2D spatial attention map (M*_s_*∈R^1×*H×W*^) are then obtained through sequential inference performed by the module. The mathematical representations of the attention process are given as:(5)$$F^{\prime }=M_{c}(F)⊗F$$
(6)$$F^{\prime \prime }=M_{s}(F^{\prime })⊗F^{\prime }$$
where the symbol ⊗ denotes an element-level multiplication in the attention process. The spatial dimension is communicated together with the channel attention levels. F″ represents the refined output.

Notably, the feature map is compressed along the spatial dimension by the channel attention mechanism to produce a one-dimensional vector. The corresponding calculation for the channel attention is expressed as:(7)$$\begin{array}{cc} M_{c}(F) & =\sigma (MLP(AvgPool(F))+MLP(MaxPool(F)))\\ & =\sigma \left(W_{1}\left(W_{0}\left(F_{\mathrm{avg}}^{c}\right)\right)+W_{1}\left(W_{0}\left(F_{\max}^{c}\right)\right)\right) \end{array}$$

The channel attention sub-module uses the shared network’s maximum and average pooling outputs to generate an attention map, as shown in Figure 4. Two distinct spatial context descriptors, referred to as F^c^_avg_ and F^c^_max_, are produced simultaneously by aggregating the spatial information of the feature maps using average pooling and max pooling. The average and maximum pooled characteristics are represented, respectively, by these two descriptors. The channel attention map M*_c_*∈R*^C^*^×1×1^ is created by feeding the two feature maps into a common network of multi-layer perceptrons (MLPs). Next, R*^C/r^*^×1×1^ is chosen as the activation value size, where r is the reduction ratio and is a sigma function. The weights W_0_∈R*^C/r^*^×*C*^ and W_1_∈R*^C^*^×*C/r*^ of the MLP are shared with the ReLU activation function that comes after W_0_.

The spatial attention mechanism compresses the channel by employing average pooling and maximum pooling in the channel dimension, which is formulated as:(8)$$\begin{array}{cc} M_{s}(F) & =\sigma \left(f^{7\times 7}([AvgPool(F);MaxPool(F)])\right)\\ & =\sigma \left(f^{7\times 7}\left(\left[F_{\mathrm{avg}}^{s};F_{\max}^{s}\right]\right)\right) \end{array}$$

To aggregate the feature data of one feature map, two pooling operations—maximum pooling and average pooling—are conducted on the channel dimension, resulting in a dual-channel feature map. Specifically, the number of maximum pooling extractions is *H × W*, and the number of average pooling extractions is also *H × W*. Consequently, two 2D feature maps are obtained; the average pooling and maximum pooling characteristics throughout the whole channel are represented by the symbols F^s^_avg_∈R^1×*H×W*^ and F^s^_max_∈R^1×*H*×*W*^, respectively. To create a 2D spatial attention map, these two maps are combined and convolved using typical convolutional layers. The convolution operation, denoted as *f*^7×7^, employs a filter size of 7 × 7.

The feature map is compressed along the spatial dimension by the channel attention mechanism to produce a one-dimensional vector. The corresponding calculation for the channel attention is expressed in Equation (7).

(2)Global Attention Mechanism

The goal of the global attention module (GAM) is to improve neural network performance by reducing the loss of useful information and boosting the representation of global interactions. A convolutional spatial attention sub-module with multi-layer perceptions and a three-dimensional channel attention sub-module are introduced to accomplish this. As shown in Figure 5, the GAM uses the channel attention mechanism and the spatial attention mechanism juxtaposition technique, similar to the CBAM approach [32]. An intermediate state F_2_ and an output F_3_ are defined as follows, given an input feature map F_1_∈R*^C×H×W^*:(9)$$F_{2}=M_{c}(F_{1})⊗F_{1}$$
(10)$$F_{3}=M_{s}(F_{2})⊗F_{2}$$

The symbols for the channel attention map and the spatial attention map are M*_c_* and M*_s_*, respectively, with element-level multiplication ⊗.

To preserve 3D information, the channel attention submodule utilizes 3D permutations. After that, it uses a two-layer multilayer perceptron (MLP) to improve the spatial and cross-dimensional relationships. The MLP is built using a compression ratio of *r*, and Figure 6 shows the channel attention submodule.

To fuse the spatial information, the spatial attention sub-module uses two convolutional layers and maintains the same compression ratio *r* as the channel attention sub-module.

By reducing feature loss and magnifying the representation of global interactions, the GAM attention mechanism improves the performance of the neural network. Here, we introduce a convolutional spatial attention submodule with multi-layer perceptron and a three-dimensional channel attention submodule. By embedding the CBAM module into the last layer of the CSP of the backbone and neck, the feature map undergoes adaptive refinement for each convolutional block of the deep network through the CBAM module. This process reduces the model’s computational complexity and establishes high-dimensional spatial features’ correlations, thereby facilitating the extraction of relevant features. The network structure incorporates the GAM and CBAM attention mechanisms, as illustrated in Table 3.

Table 3 shows the number of input source layers in the “from” column and the number of parameters in the “params” column. The “arguments” column lists information on the number of input and output channels, convolution kernel size, step size, and other relevant specifics. The “module” column lists the name of the module.

As shown in Figure 7, we used Grad-CAM to display the model’s heat map characteristics. The visualization demonstrates the requirement for the original YOLOv5s model feature extraction to be more coherent and suitable for small targets. However, after incorporating the combined attention mechanism, the model focuses on extracting critical information, reduces attention to irrelevant details, and remarkably enhances the feature extraction of small objects.

### 2.3. Bounding Box Loss Function

Object detection accuracy and effectiveness are heavily reliant on the loss function employed. Traditional object detection loss functions are based on aggregating bounding box regression metrics. However, the distance between the expected target box and the predicted box, the overlapping area, and aspect ratio are a few of the characteristics that greatly affect aggregation accuracy. Some examples include the fact that YOLOv5’s GIoU, CIoU, etc. do not take into consideration the direction discrepancy between the desired target box and the forecast box, resulting in a slower convergence speed and poorer model performance [33]. On the other hand, the SCYLLA-IoU LOSS (SIoU) [34] considers the vector’ angle between the regressed boxes and the orientation discrepancy between the anticipated box and the required item box, resulting in increased detection precision.

Conventional object detection loss algorithms are considerably improved by the SIoU loss function, as it not only considers the angle and distance between the regressed boxes, but also addresses the orientation mismatch between the predicted and desired object boxes. This improves training effectiveness and ultimately enhances target box regression’s stability, resulting in a more accurate model. The angle cost, distance cost, shape cost, and IoU cost make up the SIoU loss function.

(1)Angle cost

An extra term, LF, in the SioU loss function integrates an adaptive angle adjustment function and greatly lowers the number of variables linked to distance. As seen in Figure 8, the model first lines up the predicted box with either the X or Y axis (whichever is closest), and then it optimizes the distance along the pertinent axis.

When *α* ≤ Π/4, minimize *α*, and when *α* > Π/4, minimize *β*. The definition of LF is obtained, which can be constructed as:(11)$$\Lambda =1-2*sin^{2}\left(arcsin(x)-\frac{\pi }{4}\right),$$
where
(12)$$x=\frac{c_{h}}{\sigma }=sin(\alpha ),$$
(13)$$\sigma =\sqrt{{\left(b_{c_{x}}^{gt}-b_{c_{x}}\right)}^{2}+{\left(b_{c_{y}}^{gt}-b_{c_{y}}\right)}^{2}},$$
(14)$$c_{h}=max\left(b_{c_{y}}^{gt},b_{c_{y}}\right)-min\left(b_{c_{y}}^{gt},b_{c_{y}}\right).$$

(2)Distance cost

Based on the redefined angle cost, SIoU defines the distance cost:(15)$$\Delta =\sum_{t=x,y}\left(1-e^{-\gamma \rho_{t}}\right),$$
where
(16)$$\rho_{x}={\left(\frac{b_{c_{x}}^{gt}-b_{c_{x}}}{c_{w}}\right)}^{2},\;\rho_{y}={\left(\frac{b_{c_{y}}^{gt}-b_{c_{y}}}{c_{h}}\right)}^{2},\gamma =2-\Lambda ,$$

Equations (12) to (16) show that the effect of distance cost on the output decreases noticeably as the value of *α* approaches 0. Conversely, as *α* approaches Π/4, the impact of the distance cost on the output becomes more significant.

(3)Shape cost

A definition of the shape cost function is:(17)$$\Omega =\sum_{t=w,h}{\left(1-e^{-\omega_{t}}\right)}^{\theta },$$
where
(18)$$\omega_{w}=\frac{\left|w-w^{gt}\right|}{max\left(w,w^{gt}\right)},$$
(19)$$\omega_{h}=\frac{\left|h-h^{gt}\right|}{max\left(h,h^{gt}\right)}$$

The value of *θ* can have a variety of effects on the shape cost depending on the shape of each dataset. To ascertain the relative significance of the form cost, a certain value of *θ* is determined. A genetic algorithm is used during training to determine the ideal value of *θ* for each dataset.

(4)IoU cost

The IoU cost is described as:(20)$$IoU=\frac{\left|B\cap B^{GT}\right|}{\left|B\cup B^{GT}\right|}$$

The *L_box_* regression loss function is formulated as:(21)$$L_{box}=1-IoU+\frac{\Delta +\Omega }{2}$$

The total loss function is constructed as:(22)$$L=W_{box}L_{box}+W_{cls}L_{cls}$$

To calculate the loss function, we used a genetic algorithm to determine the values of *W_box_*_,_ *W_cls_*, and *θ*. *L_cls_* represents the focal loss, while *W_box_* and *W_cls_* are the weights for the prediction box and classification loss, respectively. Moreover, we chose a small subset from the training set and computed these values iteratively until the number of iterations was either below a threshold or the maximum number was achieved, at which time the iterations were terminated.

### 2.4. Knowledge Distillation

Knowledge distillation is a technique utilized to extract the knowledge of a large teacher model and condense it into a small student model. It can be understood as a large teacher neural network teaching his knowledge to a small student network [35,36,37].

The process is transferred from the teacher network to the student network. The teacher network is generally bloated; therefore, the teacher network provides knowledge to the student network. The student network is a relatively small network and can thus obtain a lightweight network model. Knowledge distillation adopts the teacher–student mode. In this mode, the teacher is the output party of “knowledge”, and the student is the receiver of “knowledge” [38].

The teacher has a strong learning ability and can transfer the learned knowledge to the student model with a lower learning ability, so as to improve the generalization ability of the student model. The complicated and cumbersome but easy-to-use teacher model has no upper limit; it is purely a tutor, and in reality, a simple and flexible student model is deployed. The knowledge distillation process is shown in Figure 9 below.

First, distill a deeper teacher network with a better extraction ability to obtain a logit, and distill it at *T* temperature. Then, use the classification prediction probability distribution in the Softmax layer to obtain soft targets. At the same temperature *T*, the logits in the student network are distilled, and then the category prediction probability distribution in Softmax is used to obtain the loss function *L_soft_*. Its expression is:(23)$$L_{soft}=-\sum_{j}^{N}p_{j}^{T}\mathrm{lb}\left(q_{j}^{T}\right)$$
where *C_j_* is the true label value of the *j*-th class.

Finally, *L_hard_* and *L_soft_* are weighted and summed to obtain the final loss function L. This loss function can prevent the wrong information from the teacher network from being transmitted to the student network by comparing it with the real label. In this study, the improved YOLOv5s model was used as the teacher network, and the YOLOv5s model with the large target detection layer removed by structural pruning was used as the student model for knowledge distillation to obtain the final model and reduce the amount of calculation and parameters of the improved network model.

## 3. Performance Analysis

### 3.1. Dataset and Experimental Environment

The dataset used in this study consists of 14,966 images extracted from video streams, comprising 7000 images from the public Safety Helmet Wearing and Head Detection (SHWD) dataset and 7966 images of extracted video stream frames. The images are divided into two categories: person and hat. The training set comprises 11,973 images, and the validation set comprises 2993 images, with an 8:2 ratio of training to validation data. Using two NVIDIA RTX 3060 graphics cards and the Linux operating system, the tests were carried out. Using the CUDA 11.1 computing architecture and the Pytorch deep learning framework, we built, trained, and validated our models. The batch size was 32, the workers were 8, and the image resolution was 640 × 640. The model was trained for 300 epochs with a learning rate of 0.001. The results achieved using these settings are displayed in Table 4.

### 3.2. Evaluation Criteria

As assessment measures for our model in this article, we use precision (P), recall (R), mean average precision (mAP), and detection speed (FPS). Precision and recall are defined in Equations (24) and (25).
(24)$$P=\frac{TP}{TP+FP}$$
(25)$$R=\frac{TP}{\left(TP+FN\right)}$$

True positives, or TP, in this context refers to the total number of accurately identified items. False positives, or FP, on the other hand, are the quantity of items that were mistakenly identified. Last but not least, FN stands for false negatives and denotes the quantity of items the model failed to detect. These assessment metrics offer insightful information about the model’s functionality and precision in object detection.

According to Equation (26), the average accuracy (AP) denotes the average accuracy rate under various recall rates.
(26)$$AP=\int_{0}^{1}P(R)dR$$

According to Equation (27), this yields the mean average precision (mAP).
(27)$$\;\mathrm{mAP}\;=\frac{\sum_{i=}^{N}AP_{i}}{N}$$

Here, N stands for the total number of categories, and n stands for the category.

While the IOU threshold is set to 0.5, the average AP is represented by the mAP@0.5. The average value of mAP while the IOU threshold varies from 0.5 to 0.95 in steps of 0.05 is represented by the mAP@0.5:0.95 value.

The F1-score is used to comprehensively evaluate the recall and accuracy indicators, as shown in Equation (28):(28)$$F1-\mathrm{score}=\frac{2\mathrm{TP}}{\left(\mathrm{Total}\;\mathrm{number}\;\mathrm{of}\;\mathrm{samples}+\mathrm{TP}-\mathrm{TN}\right)}$$

The number of pictures detected per second is indicated by the detection speed (FPS).

### 3.3. Ablation Experiments

To examine how various loss functions affect the YOLOv5s algorithm, we conducted experiments using commonly used loss functions, such as GIoU, CioU, DioU [39], EioU [40], and SioU. The training accuracy obtained after replacing the original GioU loss of the YOLOv5s algorithm with different loss functions is shown in Table 5.

In Table 5, we list the training accuracies of various loss functions, including GIoU, CIoU, DIoU, EIoU, and SIoU, used in the YOLOv5s algorithm. The findings demonstrate that, in comparison to other loss functions, SIoU has significantly increased the precision rate (P), recall rate (R), and mAP. Conventional object detection loss functions primarily depend on combining bounding boxes regression variables, such as the separation between the anticipated object box and the predicted box, the area that overlaps with the predicted box, and the aspect ratio. However, the use of GIoU, CIoU, etc. by YOLOv5 ignores the mismatch between the desired target frame and the prediction frame, resulting in sluggish convergence and causing the prediction frame to fluctuate throughout training. Ultimately, a poor model is produced. Using the SIoU, the vector’s angle between the regression boxes and the mismatch direction between the predicted and expected target boxes is considered, thereby changing the calculation method.

Based on the original YOLOv5s model, this study conducted ablation experiments to verify each improvement’s impact on model training. Table 6 displays the trial outcomes. Precision, recall, mAP, and mAP@0.5:0.95 of 90.3%, 86.2%, 89.8%, and 57.1%, respectively, were attained by the original YOLOv5s model. YOLOv5s-K increased recall by 0.7% and mAP by 0.7% in comparison to the original YOLOv5s algorithm, and mAP@0.5:0.95 by 1.4%. Using the K-means++ algorithm measurement method to adjust the prior frame improved the matching degree of the set target box with the preceding frame and data. From the beginning, YOLOv5s-KS significantly improved recall by 0.8% and mAP by 1.8%, compared to the YOLOv5s method. This increase is attributable to YOLOv5s-KS’s large improvement in precision, which was made possible by taking into account the vector’s angle between the regression boxes and the mismatch between the target and prediction frames while utilizing SIoU as the bounding box loss function. Comparing the upgraded model to the original YOLOv5s model, the better model saw gains in precision, recall, mAP, and mAP@0.5:0.95 of 1%, 1.1%, 2.6%, 2.1%, and 0.95, respectively. The performance of the deep neural network was enhanced by the addition of the GAM attention mechanism and the combined attention mechanism of the CSP module incorporated in the CBAM attention mechanism in the backbone and neck. This improvement was made possible by lowering the feature loss, enhancing the representation of global interactions, and adding a multi-layer three-dimensional arrangement of the channel attention sub-module and the convolutional space attention sub-module, which enhanced the efficiency of object feature extraction.

The improved YOLOv5s-KSGC model was leveraged as the teacher network, and the YOLOv5s model with the large target detection layer removed by structural pruning was utilized as the student model for knowledge distillation to obtain the final model. The experimental effect comparison between the improved YOLOv5s-Improved model and the original YOLOv5s is shown in Table 7 below. It can be seen from Table 7 that the improved model not only reduces the number of parameters and model size, but also effectively improves other indicators. Among them, mAP0.5 increased by 2.6%, mAP@0.5:0.95 increased by 2.1%, and FPS increased by 9.33.

Figure 10a–c shows the training loss of the original YOLOv5s model and the improved YOLOv5s-Improved model. Figure 10a is the Box_Loss obtained from training. It can be seen from Figure 10a that the Box_Loss of the improved model is much lower than the loss of the original YOLOv5s model training. It can be seen from Figure 10b that the Cls_Loss of the original YOLOv5s model fluctuates greatly, and the improved model significantly improves the fluctuation of Cls_Loss and reduces the loss value. It can be seen from Figure 10c that the Obj_Loss of the improved YOLOv5s-Improved model is also lower than the original YOLOv5s model at the beginning, and finally tends to be equal. The experiments prove that the improved model is reliable and stable and has higher robustness.

As can be seen from Figure 11, although the YOLOv5s-Improved and YOLOv5 training effect is good, both demonstrate the overfitting and underfitting phenomena. However, the modified model greatly increased the average accuracy in comparison to the old model, proving the viability of the revised technique.

### 3.4. Comparison of Different Methods

The YOLOv5s-Improved model was trained on the dataset to assess the performance of the suggested approach, and the outcomes were compared with those of other cutting-edge object identification models, such as SSD, Faster-RCNN, YOLOv3, ML-YOLOv3 [41], YOLOv4, YOLOv5s, YOLOv5u, YOLOv5s-DM [42], YOLOv6s, and YOLOv7-w6 [43]. Table 8 displays the experimental findings.

It can be seen from Table 8 that the improved YOLOv5s model has significantly improved mAP and FPS compared with the SSD model, Fast-RCNN model, and Faster-RCNN model under the premise of maintaining a light weight. Although the model mAP proposed in this study is similar to the YOLOv3 model and YOLOv4 model, their parameter quantity and model size are much larger than the model proposed in this paper, and the detection speed is much lower than the model in this paper. When comparing YOLOv6s, which is also a lightweight model, the parameter quantity and model size of the model in this paper are lower than YOLOv6s, and the detection accuracy speed is also higher than that of the YOLOv6s model. Due to the large number of parameters of the YOLOv7 model, this paper chose the YOLOv7-w6 model with fewer parameters for experimental comparison. It is proved by the experiments that the parameter quantity of this method is lower than that of the YOLOv7-w6 model, the detection accuracy is slightly higher than that of the YOLOv7-w6 model, and the detection speed is much higher than that of the YOLOv7-w6 model. Additionally, the method in this paper scores higher than other methods in precision, recall, and the F1-score. Compared with the ML-YOLOv3 model, the method in this paper not only has obvious advantages in precision, recall, the F1-score, and the map, but the model size and parameters are also lower than the ML-YOLOv3 model, and the detection speed is also higher than the ML-YOLOv3 model. Compared with the anchor-free YOLOv5u model, it is not as effective as the original YOLOv5s model on the helmet dataset. The advantages of the method in this paper are also superior to the YOLOv5s-DM model in precision, recall, F1-score, map, model size, and parameter quantity. Compared with the original YOLOv5s model, the results of the method in this paper show that the four indicators of precision (P), recall (R), mAP, and detection speed FPS are better than the original YOLOv5s, and have higher accuracy and detection speed. This reflects the excellent performance of the method in this paper.

## 4. Case Analysis

In this study, the proposed method was practically applied to detect helmets in various indoor and outdoor scenes at different distances. Furthermore, the detection results were compared with those obtained from the SSD, Faster R-CNN, YOLOv5s, YOLOv6s, YOLOv7-w6, and YOLOv5s-Improved models.

The results of the six approaches’ detection in a scenario with sunshine are shown in Figure 12. The detection results show that the suggested approach can identify two types of small target items in a bright outdoor setting. Figure 12a in particular illustrates the SSD model’s detection impact, with detected confidence values for the hat and person being 0.74 and 0.80, respectively. Similarly, Figure 12a,b shows the Faster R-CNN model’s detection impact, with detected confidence values of 0.75 and 0.82 for the hat and person, respectively. Figure 12c depicts the YOLOv5s model’s detection impact, with detected confidence values for the hat and person of 0.76 and 0.85, respectively. Figure 12d shows the YOLOv6s model’s detection effect, with detected confidence values of 0.77 and 0.86 for the hat and person, respectively. Figure 12e illustrates the YOLOv7-w6 model’s detection impact, with detected confidence values for the hat and person of 0.78 and 0.88, respectively. The detection impact of the suggested model is finally shown in Figure 12f, where the detected confidence values for the hat and person are 0.81 and 0.92, respectively. The detection findings indicate that the suggested approach, when used in an outdoor setting sunshine, provides much greater detection accuracy than previous target detection methods.

Figure 13 shows the detection results of the six methods in outdoor shaded scenes. The results show that two classes of small target objects are detected in the outdoor shadow environment. Figure 13a depicts the SSD model’s detection effect, and the detection’s confidence scores are 0.73 for the hat and 0.78 for the person. Figure 13b illustrates the Faster R-CNN model’s detection impact, and the detection’s confidence scores are 0.74 for hats and 0.79 for people. Similarly, Figure 13b,c displays the YOLOv5s model’s detection impact, with the detected confidence values of 0.75 for hats and 0.80 for people. Additionally, Figure 13d demonstrates the YOLOv6s model’s detection impact. The detected confidence is 0.77 for the hat and 0.82 for the person. In contrast, 13e displays the YOLOv7-w6 model’s detection effect, and the detection’s confidence scores are 0.79 for hats and 0.84 for people. The confidence gained by the detection is 0.83 for the hat and 0.85 for the person, and Figure 13f demonstrates the detection impact of the suggested approach. One may draw the conclusion that the suggested technique outperforms previous target identification algorithms in the outdoor shadow environment, leading to increased detection accuracy.

Figure 14 shows the detection results of the six methods in indoor scenarios. It can be seen from the detection results in Figure 14 that in the indoor environment, two types of small target objects are detected. Figure 14a is the detection effect of the SSD model, and the confidence obtained by the detection is 0.89 for hats and 0.70 for people. Figure 14b is the detection effect of the Faster R-CNN model, and the confidence obtained by the detection is 0.92 for the hat and 0.71 for the person. In Figure 14c, we show the detection effect of the YOLOv5s model, and the detected confidence for hats is 0.92 and 0.72 for people. Figure 14d shows the detection effect of the YOLOv6s model, and the detected confidence is 0.93 for hats and 0.73 for people. Figure 14e is the detection effect of the YOLOv7-w6 model, and the confidence obtained by the detection is 0.93 for the hat and 0.74 for the person. Figure 14f is the detection effect of the model in this paper, and the confidence obtained by the detection is 0.95 for hats and 0.80 for people. It is concluded that the detection accuracy of this method is higher than that of other target detection algorithms in different indoor and outdoor environments, which proves the feasibility and effectiveness of the improvement. Due to the improvement of the original anchor box mechanism of YOLOv5, the matching degree between the preselected box and the target box has been increased. The attention mechanism has been added to increase the extraction of effective target information features. The loss function has been improved to effectively increase the speed of prediction box regression and precision. The experiments have proved that the method in this paper can be applied to helmet detection in various scenarios, and the detection accuracy has reached more than 90%. The higher the detection accuracy, the higher the detection efficiency in actual deployment. Finally, knowledge distillation is used to reduce the number of parameters and the model size and increase the detection speed, and is more conducive to the deployment of the model.

## 5. Conclusions

In this paper, we proposed a modified YOLOv5 network to adaptively adjust the anchor box to increase the matching degree between the anchor box and the target box, which can extract discriminative image features from small targets. In the proposed method, the GAM attention mechanism is combined with the CPS module of the CBAM attention mechanism. It is added to the backbone network (Backbone) and neck network (Neck) of the original YOLOv5s network to improve the performance of the neural network by reducing the loss of feature information and amplifying the global interaction. This article introduces a three-dimensionally arranged channel attention and convolutional spatial attention sub-module with a multi-layer perceptron, and the feature map adaptively refines each convolutional block of the network structure through a combination module, which is conducive to the establishment of high dimensional spatial feature correlation and the extraction of effective features of the target. While introducing the attention mechanism, the latest SIoU LOSS is used as the bounding box loss function at the output end, which effectively improves the speed and accuracy of the prediction box regression. The experiments prove that the improved network structure has higher performance. Finally, knowledge distillation is used to realize a lightweight network to obtain the final model, which reduces the amount of computation and parameters of the improved network model and improves the detection speed FPS, which is more conducive to the deployment of the model.

According to the experimental findings, the suggested strategy enhances the accuracy indicators and mean average precision (mAP) acquired from training on the hard hat dataset. Further evidence that our technique may greatly increase the detection accuracy of small targets while meeting real-time detection requirements is shown by the large improvement in the confidence level attained by actual detection.

## Figures and Tables

**Figure 1 sensors-23-05824-f001:**
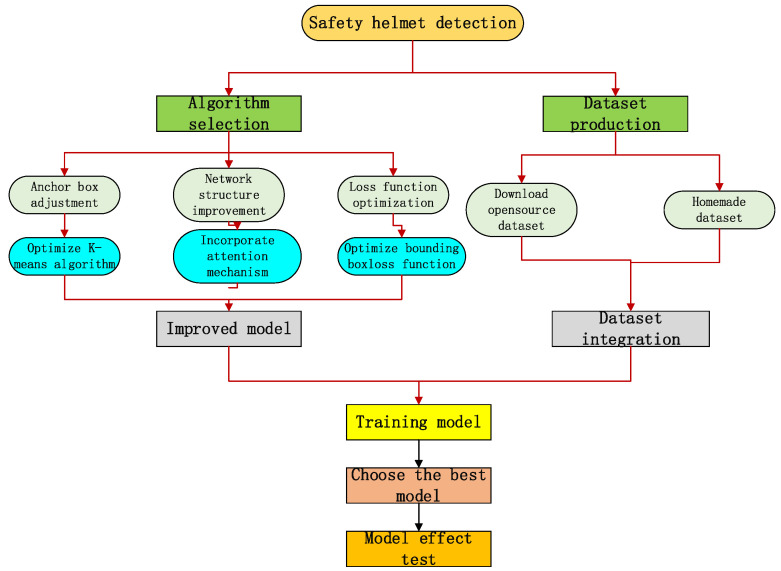
Overall framework diagram of the proposed method.

**Figure 2 sensors-23-05824-f002:**
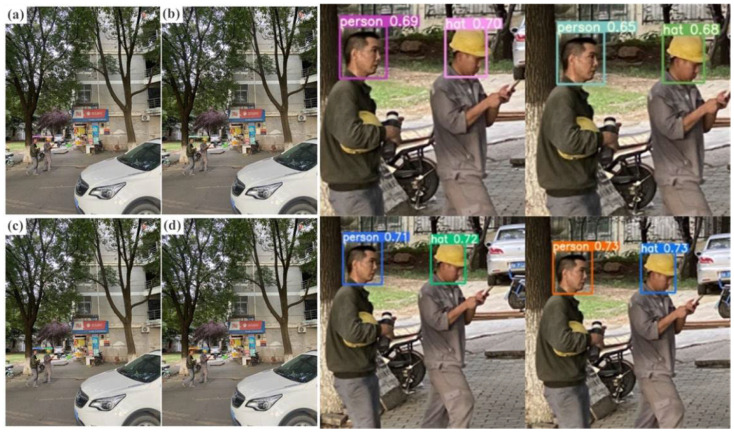
Different target detection algorithms are used for small targets with complex backgrounds in real images and actual detection effect pictures. (**a**) The detection effect diagram of the SSD model. (**b**) The detection effect diagram of the Fast R-CNN model. (**c**) The detection effect diagram of the Faster R-CNN model. (**d**) The detection effect diagram of the YOLOv5s model.

**Figure 3 sensors-23-05824-f003:**
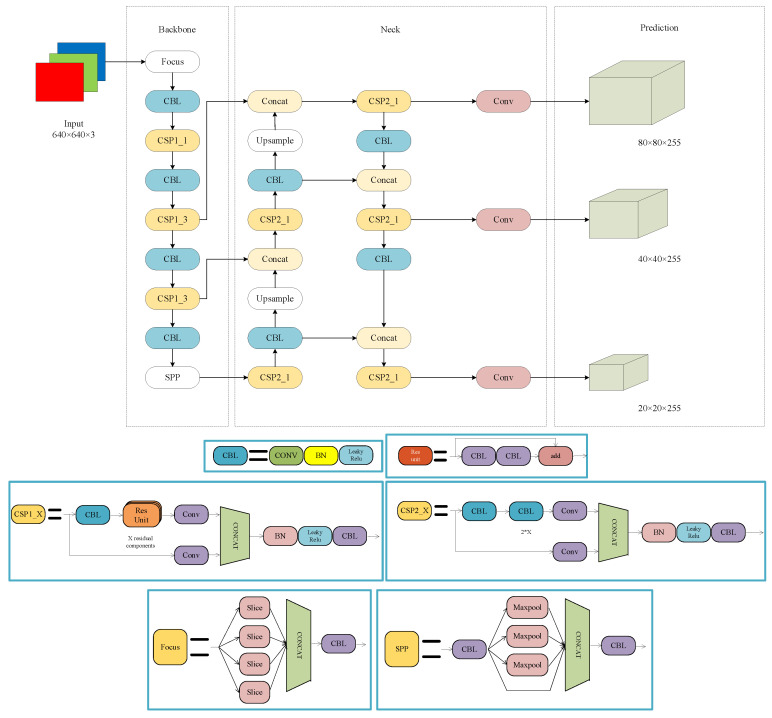
YOLOv5s network structure.

**Figure 4 sensors-23-05824-f004:**
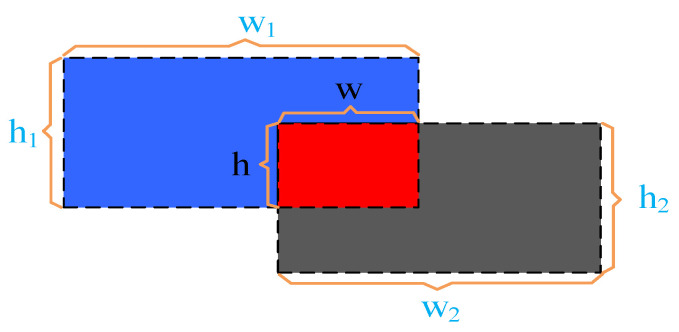
Intersection and comparison schematic diagram.

**Figure 5 sensors-23-05824-f005:**
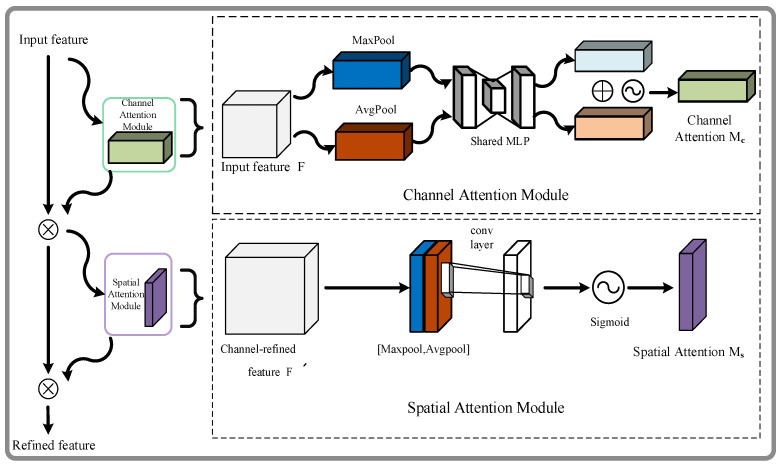
CBAM attention module.

**Figure 6 sensors-23-05824-f006:**
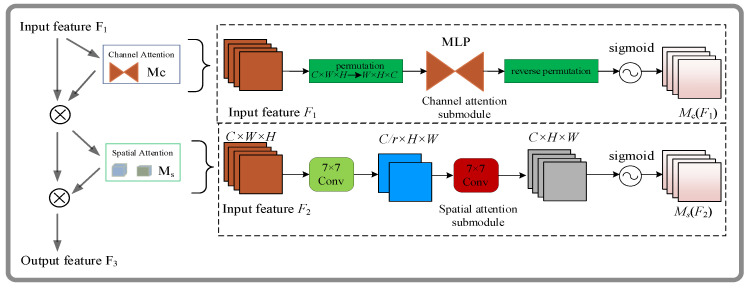
GAM attention module in our proposed method.

**Figure 7 sensors-23-05824-f007:**
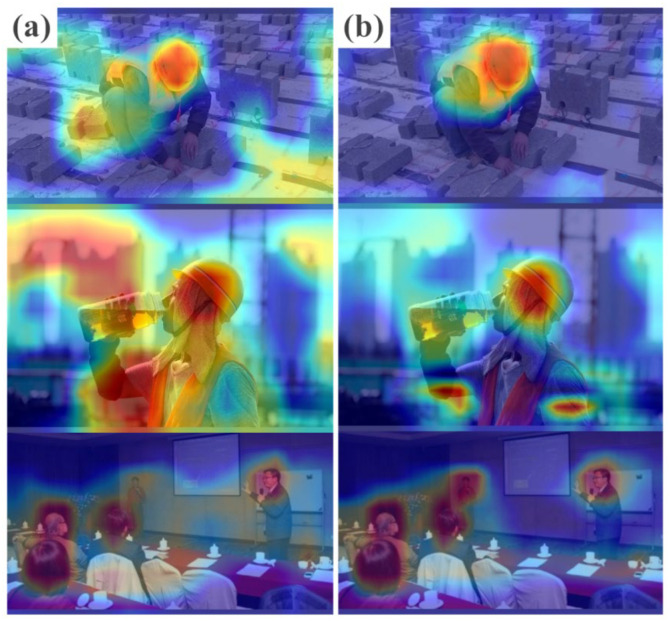
Comparison of model thermal map feature visualization output before and after adding attention mechanism. (**a**) Original YOLOv5s model. (**b**) Model after adding the combined attention mechanism.

**Figure 8 sensors-23-05824-f008:**
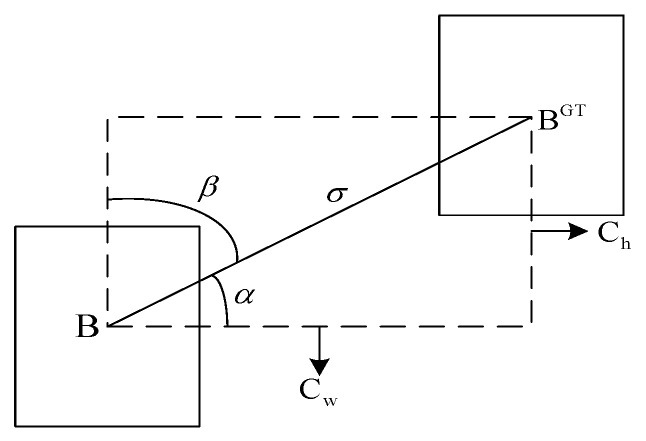
Effect of the angular factors on the loss function.

**Figure 9 sensors-23-05824-f009:**
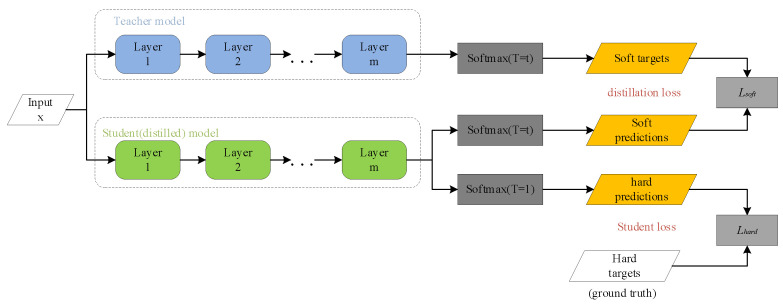
Schematic diagram of knowledge distillation process.

**Figure 10 sensors-23-05824-f010:**
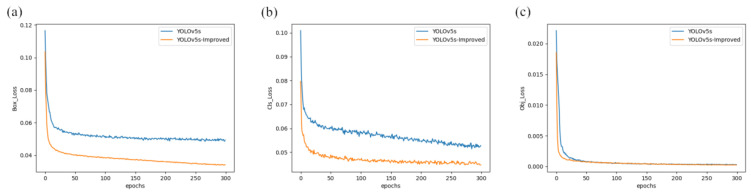
Loss comparison between this paper’s methodology with YOLOv5s. (**a**) Box_Loss comparison diagram of training results. (**b**) Cls_Loss comparison diagram of training results. (**c**) Obj_Loss comparison diagram of training results.

**Figure 11 sensors-23-05824-f011:**
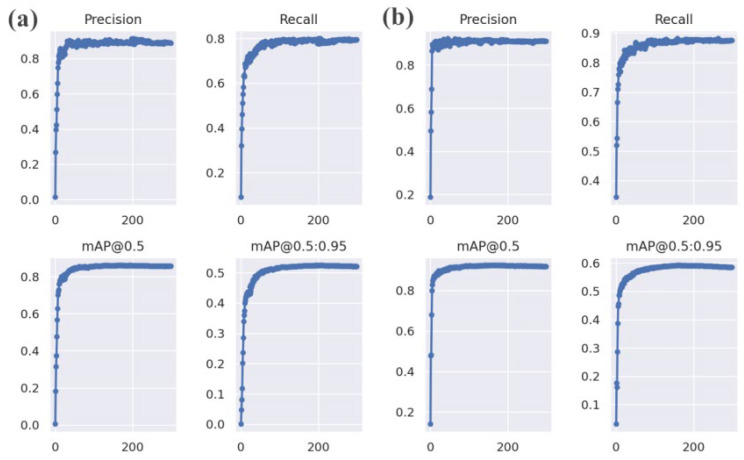
Training results for the different methods. (**a**) YOLOv5s model. (**b**) YOLOv5s-Improved model.

**Figure 12 sensors-23-05824-f012:**
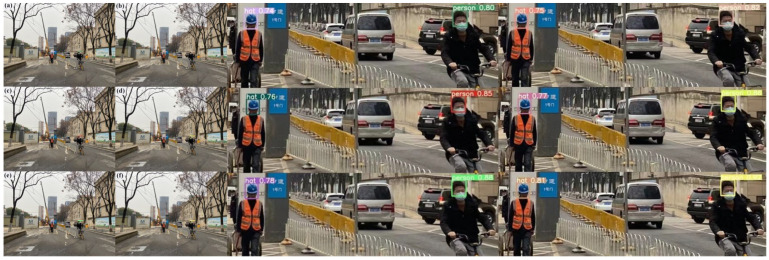
Schematic comparison of actual detection effect of different detection algorithms in outdoor environment. (**a**) The detection effect diagram of the SSD model. (**b**) The detection effect diagram of the Faster R-CNN model. (**c**) The detection effect diagram of the YOLOv5s model. (**d**) The detection effect diagram of the YOLOv6s model. (**e**) The YOLOv7-w6 model detection effect diagram. (**f**) The YOLOv5s-Improved model detection effect diagram.

**Figure 13 sensors-23-05824-f013:**
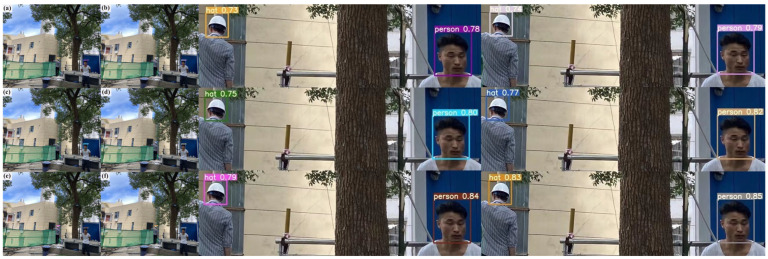
Comparison of different detection algorithms’ real detection results in darkened situations. (**a**) The detection effect diagram of the SSD model. (**b**) The detection effect diagram of the Faster R-CNN model. (**c**) The detection effect diagram of the YOLOv5s model. (**d**) The detection effect diagram of the YOLOv6s model. (**e**) The YOLOv7-w6 model detection effect diagram. (**f**) The YOLOv5s-Improved model detection effect diagram.

**Figure 14 sensors-23-05824-f014:**
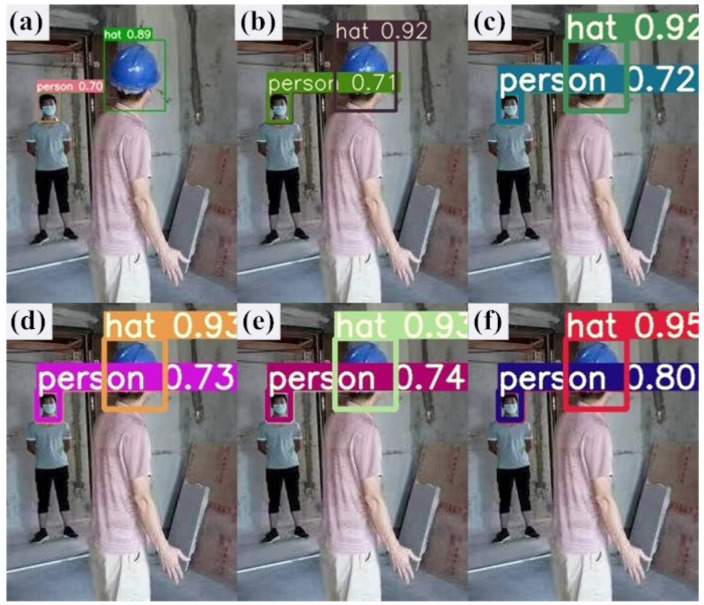
Comparison of the actual detection effect of different detection algorithms in indoor environment. (**a**) The detection effect diagram of the SSD model. (**b**) The detection effect diagram of the Faster R-CNN model. (**c**) The detection effect diagram of the YOLOv5s model. (**d**) The detection effect diagram of the YOLOv6s model. (**e**) The YOLOv7-w6 model detection effect diagram. (**f**) The YOLOv5s-Improved model detection effect diagram.

**Table 1 sensors-23-05824-t001:** A priori box distribution following K-means clustering.

Feature Map Size	Anchor Frame Size		
80 × 80	(8, 10)	(11, 13)	(15, 18)
40 × 40	(24, 27)	(36, 42)	(54, 64)
20 × 20	(87, 101)	(155, 183)	(267, 339)

**Table 2 sensors-23-05824-t002:** Algorithm clustering of the anchor block steps of K-means++.

Feature Map Size	Anchor Frame Size		
80 × 80	(8, 10)	(10, 12)	(15, 18)
40 × 40	(15, 18)	(21, 24)	(29, 33)
20 × 20	(42, 48)	(67, 77)	(120, 143)

**Table 3 sensors-23-05824-t003:** Improved YOLOv5s network structure diagram.

Number	From	Params	Module	Arguments
0	−1	3520	Focus	[3, 32, 3]
1	−1	18,560	Conv	[32, 64, 3, 2]
2	−1	18,816	C3	[64, 64, 1]
3	−1	73,984	Conv	[64, 128, 3, 2]
4	−1	156,928	C3	[128, 128, 3]
5	−1	295,424	Conv	[128, 256, 3, 2]
6	−1	625,152	C3	[256, 256, 3]
7	−1	1,639,680	GAM Attention	[256, 256]
8	−1	1,180,672	Conv	[256, 512, 3, 2]
9	−1	656,896	SPP	[512, 512, [5, 9, 13]]
10	−1	1,215,586	CBAMC3	[512, 512, 1, False]
11	−1	131,584	Conv	[512, 256, 1, 1]
12	−1	0	Upsample	[None, 2, ‘nearest’]
13	[−1, 6]	0	Concat	[1]
14	−1	361,984	C3	[512, 256, 1, False]
15	−1	33,024	Conv	[256, 128, 1, 1]
16	−1	0	Upsample	[None, 2, ‘nearest’]
17	[−1, 4]	0	Concat	[1]
18	−1	90,880	C3	[256, 128, 1, False]
19	−1	147,712	Conv	[128, 128, 3, 2]
20	[−1, 15]	0	Concat	[1]
21	−1	296,448	C3	[256, 256, 1, False]
22	−1	1,639,680	GAM Attention	[256, 256]
23	−1	590,336	Conv	[256, 256, 3, 2]
24	[−1, 11]	0	Concat	[1]
25	−1	1,215,586	CBAMC3	[512, 512, 1, False]

**Table 4 sensors-23-05824-t004:** Experimental parameters.

Parameter	Value
Lr0	0.01
Lrf	0.2
Warmup_epochs	3
Batchsize	32

**Table 5 sensors-23-05824-t005:** Comparison of experimental results with different loss functions.

Loss Function	Precision (P)/%	Recall (R)/%	mAP0.5/%	mAP0.5:0.95/%
GIoU	0.903	0.862	0.898	0.571
CIoU	0.906	0.865	0.908	0.58
DIoU	0.891	0.866	0.904	0.575
EIoU	0.891	0.868	0.91	0.582
SIoU	0.906	0.876	0.913	0.586

**Table 6 sensors-23-05824-t006:** Comparison of the ablation experiment results.

Model	K-Means++	SIoU	GAM	CBAM	Precision/%	Recall/%	mAP0.5/%	mAP0.5:0.95/%
YOLOv5s	×	×	×	×	90.3	86.2	89.8	57.1
YOLOv5s-K	√	×	×	×	89.7	86.9	90.5	58.5
YOLOv5s-KS	√	√	×	×	90.6	87.0	91.6	58.5
YOLOv5s-KSG	√	√	√	×	89.9	87.5	92.0	58.9
YOLOv5s-KSGC	√	√	√	√	91.3	87.3	92.4	59.2

**Table 7 sensors-23-05824-t007:** Comparison of the experimental effects before and after the improvement.

Model	Image Size	Params/MB	Model Size/MB	mAP0.5/%	mAP@0.5:0.95/%	FPS
YOLOv5s	640 × 640	7.06	14.4	89.8	57.1	133.33
YOLOv5s-Improved	640 × 640	5.06	12.5	92.4	59.2	142.66

**Table 8 sensors-23-05824-t008:** Comparison table of training results of different algorithms.

Model	Image Size	Params/MB	Model Size/MB	P/%	R/%	F1-Score/%	mAP0.5/%	mAP@0.5:0.95/%	FPS
SSD	512 × 512	41.18	200	83.5	78.9	41.8	77.5	/	48.5
Faster-RCNN	1000 × 600	60.17	420	87.5	82.3	44.5	82.6	/	11.6
YOLOv3	640 × 640	61.95	123.5	86.5	84.5	46.8	92.3	60.7	37.59
ML-YOLOv3	640 × 640	18.15	37	89.5	85.8	47.5	90.2	58.2	88.88
YOLOv4	640 × 640	52.92	112.6	87.2	85.1	47.4	91.5	58.2	45.3
YOLOv5s	640 × 640	7.06	14.4	90.3	86.2	48.2	89.8	57.1	133.33
YOLOv5u	640 × 640	6.52	11.4	89.8	86.2	47.5	88.5	56.2	136.5
YOLOv5s-DM	640 × 640	7.06	14.4	90.5	86.1	48.8	90.2	56.5	133.33
YOLOv6s	640 × 640	/	38.16	89.5	85.8	48.3	90.9	57.9	79
YOLOv7-w6	640 × 640	69.83	140.1	90.1	87.2	48.5	91.7	58.8	55.4
YOLOv5s-Improved	640 × 640	5.06	12.5	91.2	87.1	51.5	92.4	59.2	142.66

## Data Availability

The data used to support the findings of this study are available from the corresponding author upon request.

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
