# Peer review of "Research on Safety Helmet Detection Algorithm Based on Improved YOLOv5s"

_sensors, 2023, doi:10.3390/s23135824_

Round 1
Reviewer 1 Report
The paper describes the detection of safety helmet with YOLOv5s architecture.
The architecture YOLOv5s has been selected from a set of architectures. The optimization of the algorithm is indicated in three points, and there is not comparison with the other YOLO architectures or a motivation for the choice of this specific architecture.
The k-mean and kmeans++ are chosen to select the most promising areas. For this point, some other detail should be given. Table 1 and Table 2 collect frame size, but some other detail and clarification should be given: are these values empirical or are they chosen out of an optimization?
The blocks CBAM (convolutional block attention module) and the GAM (global attention module). A figure of the entire architecture, showing how these blocks are applied, is missing.
In figure 4, the round with a tilde inside is not specified, and makes the description unclear.
The description of features, processing, etc are not coherent with other images inside the paper.
The result of the experimental part is shown in Section 3.
Tables 6 and 7 should be improved for readability and better formatted.
In images 10,11 and 12 are shown the results of the image. The caption should be enriched with the technique that is applied to each of them.
In table 7, the unit of measurement should be better clarified. The analysis of the results according to the parameters, performance and processing speed should be better explained, and a in-depth analysis should be given.
Minor issues:
Row 257, page 7 has a word that is incomplete (…ably, the feature map)
Figure 4 has different fonts inside the image, all the text should be uniform
Row 443 and 445, page 13, show works that are wrongly separated between two lines.
Some sentence should be checked and some detail must be provided. In general the paper is clearly written although some point is obscure
Reviewer 2 Report
This work addresses and improvement of the YOLOv5s network, a lightweight deep learning-based object identification network model, applied to a helmet detection algorithm that aims at enhancing security of work environments. The topic is interesting for health and safety processes at applied engineering environments. I suggest to address the following issues:
11. There is a lack of connection with the special issue "Human-Robot Interaction for Intelligent Education and Engineering Applications", since the introduction is too short and only a few references are addressed. The authors should expand this section, with more referents in order to show how they work is actually advancing the state of the art with this contribution.
22. English language should be revised within the document.
33. The discussion of results is too narrow and only limited to the numerical part. But, how these results are connected to the state of the art?
44. The conclusions section should also be enhanced after proving a better state of the art and a better discussion of the results
English language should be revised within the document.
Reviewer 3 Report
This paper proposed a modified version of YOLOV5 network, a lightweight deep learning based object identification model to target safety helmets detection. The modified model optimizes YOLOV5 algorithm and integrates specific improvements, including anchor box adjustment, network structure improvement and bounding box loss function optimization. Evaluations demonstrate that the modified YOLOV5 model increases detection accuracy and reduces training time compared to prior works.
Overall, the work is interesting and targets a meaning problem. Also, the design is detailed, and evaluations are comprehensive. However, there are some comments below and it is advised to address them.
First, is there any chance that the work adopts the anchor-free models in the design instead of anchor-based YOLOV5?
Second, the works claims that the modified model reduces training time, however, it appears the work does not discuss the point in detail. It is advised to justify it.
Third, as the model takes into model complexity and aims at lightweight model, is there any chance that the work properly discuss the deployment performance of proposed model, like memory, computation cost in practice?
A few typos can be fixed.
Reviewer 4 Report
The research paper focuses on the problem of detecting whether construction workers are wearing safety helmets to prevent accidents and injuries. the paper focuses on the improvement of the adaptive anchor frame mechanism in YOLOv5 using the K-means++ algorithm. By combining K-means++ and Intersection-over-Union (IOU) distance metric, the proposed method generates prior bounding boxes with higher IOU values, improving object recognition accuracy. The experiments conducted by the authors demonstrate the effectiveness of the approach for helmet detection. Based on the recent literature, please find the following findings.
Strengths:
- The performance of the proposed YOLOv5s-KSGC model is compared with other state-of-the-art object detection models, such as Faster-RCNN, SSD, YOLOv3, YOLOv4, YOLOv6s, YOLOv7, and YOLOv7-w6. The comparison shows that the proposed model outperforms SSD and Faster-RCNN in terms of mAP and FPS while maintaining a lightweight architecture.
Areas of improvement:
- The description provided does not offer a detailed explanation of the loss functions (GIoU, CIoU, DIoU, EIoU, SIoU) and the specific improvements made to the YOLOv5s model. Without further clarification, it is difficult to assess the technical aspects and understand the exact modifications made.
- While the research presents training accuracies and mean average precision (mAP) as evaluation metrics, other important metrics such as precision, recall, and F1 score are not reported. These additional metrics would provide a more comprehensive assessment of the proposed method's performance.
- While the research compares the proposed approach with several existing models, it does not include comparisons with more recent state-of-the-art models like YOLO-NAS, and [1-3] .
- The paper mentions conducting ablation experiments to evaluate the impact of each enhancement, but it is not clear if the paper provides detailed results or analysis of these experiments. A more comprehensive ablation study would help in understanding the individual contributions and importance of each modification.
- Factors such as computational efficiency, model size, and the impact of hardware constraints are not adequately addressed, limiting the applicability of the proposed approach in practical scenarios.
To gain a thorough understanding of the research paper's contributions and potential implications, it is crucial to take into account the limitations alongside its notable strengths.
References:
[1] Zhou, Fangbo, Huailin Zhao, and Zhen Nie. "Safety helmet detection based on YOLOv5." In 2021 IEEE International conference on power electronics, computer applications (ICPECA), pp. 6-11. IEEE, 2021.
[2] Tan, Shilei, Gonglin Lu, Ziqiang Jiang, and Li Huang. "Improved YOLOv5 network model and application in safety helmet detection." In 2021 IEEE International Conference on Intelligence and Safety for Robotics (ISR), pp. 330-333. IEEE, 2021.
[3] Deng, Lixia, Hongquan Li, Haiying Liu, and Jason Gu. "A lightweight YOLOv3 algorithm used for safety helmet detection." Scientific reports 12, no. 1 (2022): 10981.
Round 2
Reviewer 2 Report
The authors have improved the manuscript. Some English language revisions are still needed.
Some English language revisions are still needed.
Author Response
I have modified it.Please see the attachment.
